# Effect of Hydroxyapatite/β-Tricalcium Phosphate on Osseointegration after Implantation into Mouse Maxilla

**DOI:** 10.3390/ijms24043124

**Published:** 2023-02-04

**Authors:** Sanako Makishi, Taisuke Watanabe, Kotaro Saito, Hayato Ohshima

**Affiliations:** Division of Anatomy and Cell Biology of the Hard Tissue, Department of Tissue Regeneration and Reconstruction, Niigata University Graduate School of Medical and Dental Sciences, Niigata 951-8514, Japan

**Keywords:** dental implants, hydroxyapatite, maxilla, osseointegration, osteopontin, titanium, tooth extraction

## Abstract

In our previous study we established an animal model for immediately placed implants using mice and clarified that there were no significant differences in the chronological healing process at the bone-implant interface between immediately and delayed placed implants blasted with hydroxyapatite (HA)/β-tricalcium phosphate (β-TCP) (ratio 1:4). This study aimed to analyze the effects of HA/β-TCP on osseointegration at the bone-implant interface after immediately placed implants in the maxillae of 4-week-old mice. Right maxillary first molars were extracted and cavities were prepared with a drill and titanium implants, blasted with or without HA/β-TCP, were placed. The fixation was followed-up at 1, 5, 7, 14, and 28 days after implantation, and the decalcified samples were embedded in paraffin and prepared sections were processed for immunohistochemistry using anti-osteopontin (OPN) and Ki67 antibodies, and tartrate-resistant acid phosphatase histochemistry. The undecalcified sample elements were quantitatively analyzed by an electron probe microanalyzer. Bone formation occurred on the preexisting bone surface (indirect osteogenesis) and on the implant surface (direct osteogenesis), indicating that osseointegration was achieved until 4 weeks post-operation in both of the groups. In the non-blasted group, the OPN immunoreactivity at the bone-implant interface was significantly decreased compared with the blasted group at week 2 and 4, as well as the rate of direct osteogenesis at week 4. These results suggest that the lack of HA/β-TCP on the implant surface affects the OPN immunoreactivity on the bone-implant interface, resulting in decreased direct osteogenesis following immediately placed titanium implants.

## 1. Introduction

Clinicians and researchers who are involved with dental implants require a comprehensive understanding of the biology of wound healing after endosseous implant placement. Osseointegration, as defined by Brånemark, refers to the direct contact between living bone and the implant surface at the level of a light microscope [1] and is considered to be a determinant of successful implant therapy. Mesenchymal stem cells differentiate into pre-, immature-, and mature-osteoblasts, and runt-related transcription factor 2 (Runx2), SP7, and Wnt signaling act as promoting factors of this differentiation process [2]. Osteoclast progenitor cells differentiate into osteoclasts via the macrophage colony-stimulating factor (M-CSF) receptor, receptor activator of nuclear factor-κ B (RANK) through the secretion of M-CSF and RANK ligand expression by osteoblasts that are activated by 1,25(OH)_2_D_3_ and parathyroid hormone (PTH), and these activated osteoclasts resorb bone [3]. The healing of peri-implant tissues can be divided into the bioreactive (bleeding, hemostasis, blood clots, and blood clot degradation), osteoconductive (migration and adhesion of mesenchymal cells and osteoblast differentiation), bone-forming (collagen formation by osteoblasts, calcification, and bone formation), and bone-remodeling (bone resorption and apposition) phases [4]. In the mouse experimental model [5], the bioreactive (0–3 days), osteoconductive (3–5 days), neoplastic (5–7 days), and bone remodeling (2–4 weeks) phases occur earlier than the bone-forming (1–2 weeks) and bone remodeling (6 weeks) phases in humans. Osseointegration can be divided into two modalities [4]: distance osteogenesis, in which bone is gradually added from the surrounding preexisting bone to the implant, and contact osteogenesis, in which mesenchymal stem cells migrate onto the implant surface and differentiate into osteoblasts to form bone. We have previously proposed that the former is called indirect osteogenesis and the latter is called direct osteogenesis [6]. Our previous study established an experimental animal model for the immediate implant placement using a titanium implant that was blasted with hydroxyapatite (HA)/β-tricalcium phosphate (TCP) in the mouse maxilla [5]. In this model, indirect and direct osteogeneses occur simultaneously. In indirect osteogenesis, osteoclast precursor cells are recruited onto the damaged bone surface and differentiate into bone-resorbing osteoclasts. When the osteoclasts leave the bone surface after bone resorption, osteoblast progenitors are recruited onto the bone surface where they differentiate and become osteocytes after bone formation. Although bone deposition continuously occurs from the preexisting bone to the implant surface, eventually establishing osseointegration, the intervening cells remain and are embedded in the matrix at the bone-implant interface. In direct osteogenesis, osteoclast precursor cells are recruited onto the implant surface where they differentiate and become polarized. When the osteoclast-like cells leave the implant surface, osteoblast precursor cells are recruited onto the implant surface where they differentiate into osteoblasts and deposit cell-free bone matrix at the bone-implant interface. Bone deposition continues from the implant surface to the preexisting bone. Thus, promoting direct osteogenesis contributes to achieving the osseointegration faster.

Although osteopontin (OPN) is produced by osteoblasts, osteocytes, and osteoclasts as well as inflammatory cells in the bone, OPN functions as a bridge between HA and the extracellular matrix [7]. OPN receptors contain integrins and CD44 variants and are involved in the adhesion, migration, and survival of various cell types. During direct osteogenesis, inflammatory cells and osteoclast lineage cells produce OPN, which is deposited onto the implant surface [6]. As a result, there is a correlation between the rate of direct osteogenesis and OPN deposition on the implant surface. Furthermore, the addition of recombinant OPN protein onto the implant surface results in the early establishment of direct osteogenesis [8]. However, there is no data available regarding the factors that promote the deposition of OPN onto the implant surface.

The initial stage where osteoblasts are recruited onto the surface of the implant is defined as “osteoconduction” and the ability to form bone in the areas other than bone tissue is called “osteoinduction” [9]. Similarities of HA to the bone minerals together with the HA bioactivity and biocompatibility have made it a promising scaffold for bone tissue engineering [10]. Although HA has been widely used in bone regeneration and dental implants, the effect of HA on cellular events are not fully understood. The key questions that this study aims to answer are: (i) does HA on the implant surface promote direct osteogenesis, and if it does, (ii) how does HA affect the cellular events during the osseointegration process after implant placement. Regarding the effect of HA/β-TCP, its presence has been reported to increase the attachment level and bone regeneration in the treatment of periodontal osseous defects [11]. Direct and indirect osteogenesis occurs simultaneously, and osseointegration is completed when the two meet. If direct osteogenesis does not occur, the time to osseointegration completion is extended. Clinically, it is desirable for direct and indirect osteogenesis to occur simultaneously and for osseointegration to be established at an early stage because initial fixation of the implant is important. This study aimed to clarify the effect of HA on osseointegration at the bone-implant interface after implantation in the maxillae of mice.

## 2. Results

### 2.1. Day 1

In both the blasted (HA) and non-blasted (smooth: Sm) groups on postoperative day 1, the bone-implant interface consisted of a fibrin network and inflammatory cellular infiltrate, while the preexisting bone surface was positive for tartrate-resistant acid phosphatase (TRAP) and OPN (Figure 1).

### 2.2. Day 5

On postoperative day 5, granulation progressed at the peri-implant interface and bone formation began on the implant surface in the HA group (Figure 2a–f). In addition, TRAP-positive osteoclast lineage cells appeared around the implants, and OPN-positive reactions were observed at the bone-implant interface. In contrast, neither bone formation nor OPN-positive reaction was observed around the implant in the Sm group (Figure 2g–i).

### 2.3. Weeks 2–4

At postoperative week 2, both the HA and Sm groups showed progressive bone formation over the entire implant. In both groups, OPN-positive reactions were observed at the bone-implant interface at the site of bone formation, and TRAP-positive reactions were observed in the new bone (Figure 3a–f).

Although the histological features of postoperative week 4 were similar to those of week 2 in both the HA and Sm groups, bone formation progressed all around the implants (Figure 3g–l). In both groups, OPN-positive reactions were observed in the bone-implant interface at the site of bone formation, and TRAP-positive reactions were observed in the new bone.

The success rates of immediately placed implants in the HA and Sm groups were 93.3% (36/37) and 74.4% (29/39), respectively.

### 2.4. Osseointegration and OPN-Positive Rates and Cell Proliferation

Although the osseointegration rate of postoperative week 2–4 was not significantly different between the HA and Sm groups, there was a significant difference in direct osteogenesis between HA and Sm groups at week 4 (Figure 4a). OPN-positive rates in the HA group were significantly higher than those of the Sm group at week 2 and 4 (Figure 4b). Cell proliferative activity peaked on postoperative day 5 in both the HA and Sm groups and was significantly higher in the Sm group than in the HA group on day 5 (Figure 4c).

### 2.5. EPMA Analysis

Electron probe microanalyzer (EPMA) images of the HA and Sm groups at postoperative week 4 showed direct contact completion between the implant surface and surrounding bone, although certain areas were not covered with bone (Figure 5). Quantitative analysis showed that the phosphorus levels were significantly higher in the HA group for mineral concentrations at postoperative week 4 (Figure 6). More precise measurements of the mineral concentrations showed no significant differences between the HA and Sm groups (Figure 7). In addition, calcium deposition on the surface of the implant surface was observed in both groups.

## 3. Discussion

Bone formation occurred on the preexisting bone surface (indirect osteogenesis) and/or on the implant surface (direct osteogenesis), and osseointegration was established at postoperative week 4 in both the HA and Sm groups. A comparison of the osseointegration rates, including direct and indirect osteogenesis, between the two groups revealed that direct osteogenesis was significantly lower in the Sm group at postoperative week 4. In addition, OPN immunoreactivity at the bone-implant interface was significantly decreased in the Sm group at postoperative weeks 2 and 4. These results indicate that the lack of HA on the implant surface affected the OPN immunoreactivity at the bone-implant interface, resulting in decreased direct osteogenesis after immediate placement of titanium implants. In the experiment in which a titanium implant is placed in a rat maxilla, the healing pattern is different depending on the gap between the bone and implant [12]. Bone formation begins in an island-like fashion if there is a large gap at the bone-implant interface, whereas if there is a narrow gap preexisting bone is first resorbed by osteoclasts followed by bone formation, resulting in indirect osteogenesis. In addition, the area where the bone and implant are in close proximity may remain dead bone for a long period after implantation and eventually undergo remodeling. When HA-coated titanium implants are placed in the rat maxilla, osteoclast-like cells appear on the implant surface, followed by osteoblast alignment on the implant surface and then direct osteogenesis from the implant surface [13]. Thus, the presence of HA on the implant surface indicates that direct osteogenesis is stimulated. Furthermore, it contributed to the high success rate of immediately placed implants in the HA group (93.3%), compared with that of Sm group (74.4%). Regarding the healing of long bone diaphyseal fractures, many therapeutic strategies, such as scaffolds, growth factors, cell therapies, and systemic pharmacological treatments, have been proposed in combination with surgical treatment to enhance the healing process. Clinical evidence of scaffolds effect on bone repair of acute long bone shaft fractures revealed that the level of clinical evidence of HA or β-TCP was weak, compared with autologous or allogenic bone graft [14]. The combination of growth factors including bone morphogenetic proteins or platelet rich plasma (PRP), an autologous blood concentrate suspension of platelets, may be recommended to accelerate the healing process during HA-blasted titanium implantation. A systematic review suggests that PRP has a positive effect on secondary implant stability after implant placement in patients [15].

A problematic complication after dental implant treatment is peri-implantitis, with an incidence rate of 56% [16]. Peri-implantitis is defined as mucosal lesions with suppuration and deepened pockets and a loss of supporting marginal bone. Biofilm formation plays an important role in the initiation and progression of peri-implant diseases and is associated with gram-negative anaerobes similar to those found around natural teeth in patients with severe chronic periodontitis [17]. Machined-surface implants have been replaced with implants that utilize two technologies to improve the osteoconductivity of titanium implants. One is a technique where the metal implant is coated with bioactive compounds that promote bone formation, and the other is a technique in which a rough surface is formed directly on the metal implant surface [18]. Machined-surface implants have the highest rate of decontamination, whereas HA-coated implants have the lowest rate [19]. Thus, there is a trade-off between osteoconductivity and the risk of peri-implantitis.

This study demonstrated that the rate of OPN positivity at the bone-implant interface correlated with the rate of osseointegration. OPN plays an important role in bone remodeling, where the resting osteoblasts are activated by 1,25(OH)_2_D_3_ and PTH to secrete proteinase and OPN [7]. OPN is deposited at the calcification front of the bone matrix, and subsequently osteoclasts are recruited at the calcified bone tissue. When the osteoclasts resorb the bone matrix and secret OPN, OPN is deposited at the bone matrix in the resorbed fossa and osteoblasts are recruited to begin bone formation. During direct osteogenesis, the deposition of OPN on the implant surface also occurs prior to bone formation. Thus, the deposition of OPN on the implant surface triggers direct osteogenesis. The importance of OPN in direct osteogenesis is supported by reports that have demonstrated that direct osteogenesis is significantly reduced in *Opn*-deficient mice [6] and that the addition of recombinant OPN on the surface of implants in wild-type mice causes early direct osteogenesis [20].

In this study, EPMA analysis showed that calcium deposition on the implant surface was observed in both the HA and Sm groups. In addition, a gap existed at the bone-implant interface as seen in the back scattered electron images. When the HA and Sm implants were compared, the gap was wider in the HA implants. The gap probably included artifacts due to the width of the HA blasting and the expansion and contraction of the resin. Elemental analysis by EPMA may not accurately reflect the elemental concentrations in vivo because of edge effects at the boundaries between samples made of different materials, such as the bone-implant interface. However, a comparison of the elemental concentrations of calcium and phosphorus indicates that calcium incorporation onto the titanium implant is higher than phosphorus, suggesting that calcium deposition on the implant surface occurs in vivo as well.

## 4. Materials and Methods

### 4.1. Animals and Experimental Procedure

Male Crlj:CD1 (ICR) mice were purchased from Charles River Laboratories (Yokohama, Japan). All surgeries were conducted under anesthesia using an intraperitoneal injection of a combined solution (0.05–0.1 mL/10 g) of 1.875 mL Domitor^®^ (Nippon Zenyaku Kogyo Co., Ltd., Koriyama, Japan), 2 mL midazolam (Sandoz KK, Tokyo, Japan), 2.5 mL Vetorphale^®^ (Meiji Seika Pharma Co., Ltd., Tokyo, Japan), and 18.625 mL physiological saline.

### 4.2. Immediate Implant Placement

The right maxillary first molars (M1) from 4-week-old mice were extracted with a pair of modified dental forceps under anesthesia and replaced with titanium implants blasted with HA and β-TCP (1:4) (HA group) or non-blasted machined-surface titanium implants (Sm group). The implant design was a cylindrical, threaded screw type [5]. The detailed procedure has been reported in a previous study [5]. The surface morphology of the non-blasted implant (secondary and backscattered images) on the implant surface was analyzed using an EPMA (EPMA-1610; Shimadzu, Kyoto, Japan) (Figure 8a–c). The percentage weights of each element were as follows: titanium 84.9%, vanadium 8.6%, aluminum 5.9%, and iron 0.7% (Figure 8d). The information of the blasted implant has been detailed in a previous study [5].

### 4.3. Histological Procedure and Immunohistochemical and Histochemical Analysis

Following the fixation of mice at 1, 5, 7, 14, and 28 days after implantation (Table 1), the samples were processed for the following procedures: decalcified samples were processed for Hematoxylin & Eosin (H&E) and Azan staining, immunohistochemistry for OPN and Ki67, and TRAP histochemistry. Detailed information regarding these procedures is presented in a previous study [5].

### 4.4. EPMA Analysis

Undecalcified samples embedded in Epon 812 (Taab, Berkshire, UK) were ground down to be exposed at a position approximately equal to the central plane of the implants and were used for element analysis using an EPMA (EPMA-1610, Shimadzu Co., Kyoto, Japan). The EPMA settings were as follows: spot size 1 μm; pixel matrix 380 × 380 to 446 × 446; voltage 15.0 kv; electrical current 19.95–30.10 μA. The density of calcium, magnesium, phosphorus, and titanium in the surrounding bone between the screw pitches was analyzed.

### 4.5. Statistical Analysis

The number of Ki67-positive cells at the bone-implant interface of each specimen (283 × 355 μm^2^ grid was selected) was counted by the counter tool in Photoshop 2021 (Adobe Inc., San Jose, CA, USA). Data were obtained from 21 maxillae from the HA and Sm groups (Table 1) for the cell proliferation assay using the immunoreactivity of Ki67. The rate of OPN-positive perimeter around the implant or the direct and indirect osteogenesis was statistically analyzed in the OPN immunostained or H&E-stained sections using the two-tailed Student’s *t*-test in the same manner as our previous study [6]. The percentage of osseointegration and OPN-positive perimeters in the total perimeter of the bone-implant interface was calculated using software (Image J 1.45s; National Institutes of Health, Bethesda, MD, USA). The direct and indirect osteogeneses were determined in the histological sections: the direct osteogenesis showed the direct bone deposition on the implant surface, whereas the soft tissue intervened at the bone-implant surface in the indirect osteogenesis. Furthermore, the number of Ki67-positive cells among the different stages after implantation was compared using one-way ANOVA followed by the Bonferroni test for multiple comparisons and the rate of osseointegration, OPN-positive perimeter, and the number of Ki67-positive cells between the different groups were compared using the two-tailed Student’s *t*-test with statistical software after the confirmation of data normality and homogeneity of variance (SPSS 16.0J for Windows; SPSS Japan, Tokyo, Japan). The threshold for significance was defined as α = 0.05. The samples that did not demonstrate a normal distribution were compared with the Kruskal–Wallis test followed by the Bonferroni test for multiple comparisons for more than three groups or the Mann–Whitney U test for two groups. Data were reported as mean + SD, *P* denoted the *p*-value.

## 5. Conclusions

In the Sm group, the OPN immunoreactivity rate at the bone-implant interface significantly decreased compared with the HA group at weeks 2 and 4, as well as the rate of direct osteogenesis at week 4. These results suggest that the presence of HA/β-TCP on the implant surface affects the OPN immunoreactivity at the bone-implant interface, resulting in the increase of direct osteogenesis following the immediately placed titanium implant and contributes to the high success rate in the HA group. A complication after dental implant treatment is peri-implantitis. Machined-surface implants have the highest rate and HA-coated implants have the lowest rate of decontamination. Thus, there is a trade-off between osteoconductivity and peri-implantitis risk.

## Figures and Tables

**Figure 1 ijms-24-03124-f001:**
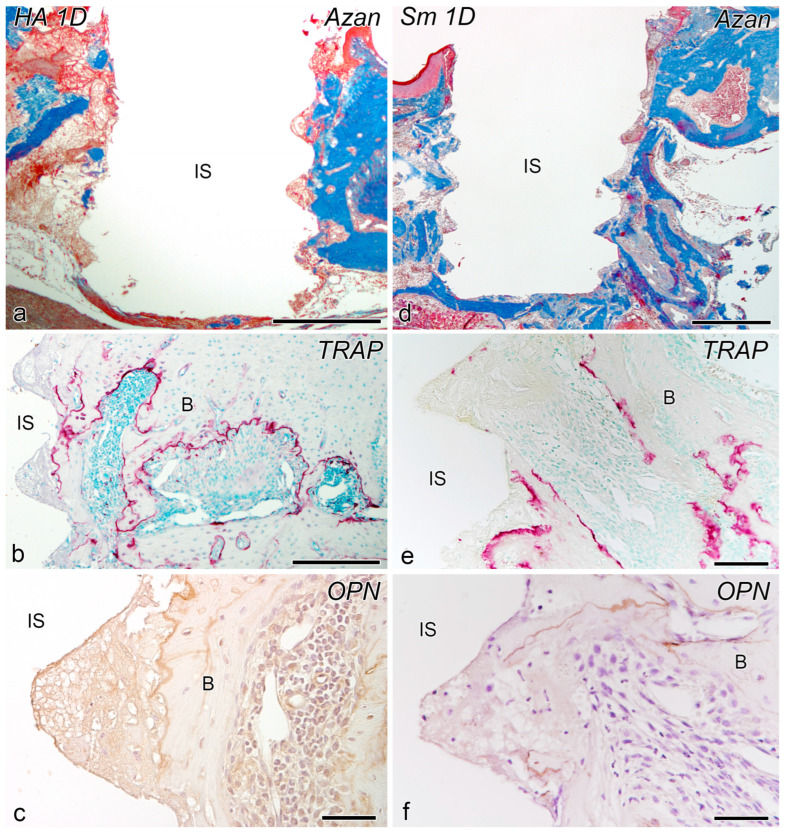
Azan staining (**a**,**d**), tartrate-resistant acid phosphatase (TRAP) reaction (**b**,**e**), and osteopontin (OPN)-immunoreactivity (**c**,**f**) in the tissues surrounding the implants in the blasted (HA) (**a**–**c**) and non-blasted (Sm) groups (**d**–**f**) at day 1 after implantation. (**a**–**f**) The bone-implant interface consists of a fibrin network and inflammatory cellular infiltrate, while the preexisting bone surface is positive for TRAP and OPN. B, bone; IS, implant space. Scale bars = (**a**,**d**) 500 μm, (**b**) 250 μm, (**e**) 100 μm, and (**c**,**f**) 50 μm.

**Figure 2 ijms-24-03124-f002:**
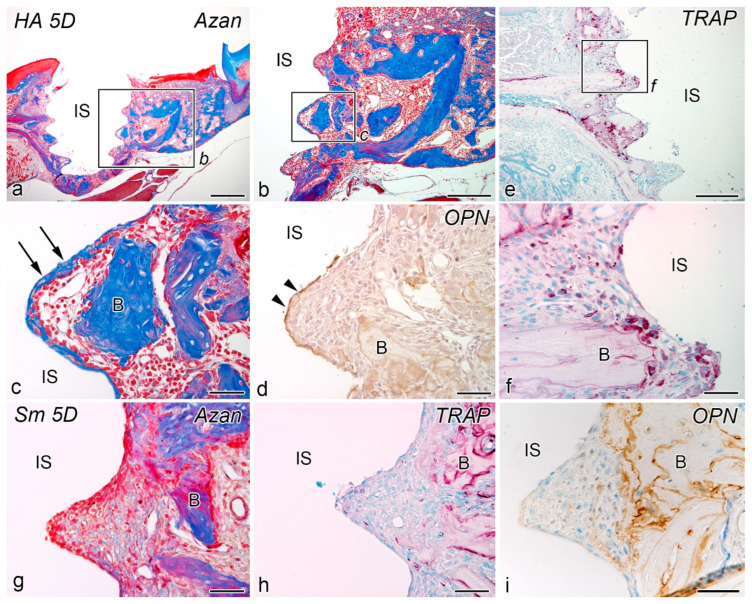
Azan staining (**a**–**c**,**g**), TRAP reaction (**e**,**f**,**h**), and OPN-immunoreactivity (**d**,**i**) in tissues surrounding the implants in the HA (**a**–**f**) and Sm groups (**g**–**i**) at day 5 after implantation. (**b**,**c**,**f**) are the higher magnifications of the boxed areas in (**a**,**b**,**e**), respectively. (**a**–**f**) Granulation progresses at the bone-implant interface and bone formation has begun on the implant surface. TRAP-positive osteoclast lineage cells appear around the implants, and an OPN-positive reaction (arrowheads) is observed at the bone-implant interface. (**g**–**i**) No bone formation is observed around the implant. Arrows, direct osteogenesis; B, bone; IS, implant space. Scale bars = (**a**) 500 μm, (**b**,**e**) 250 μm, and (**c**–**i**) 50 μm.

**Figure 3 ijms-24-03124-f003:**
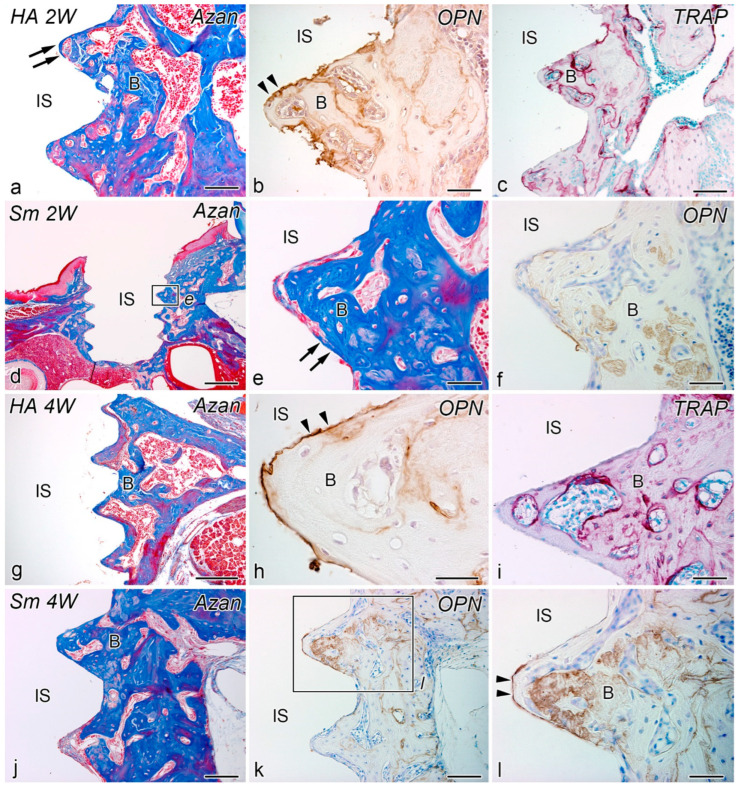
Azan staining (**a**,**d**,**e**,**g**,**j**), TRAP reaction (**c**,**i**), and OPN-immunoreactivity (**b**,**f**,**h**,**l**) in tissues surrounding the implants in the HA (**a**–**c**,**g**–**i**) and Sm groups (**d**–**f**,**j**–**l**) at week 2 (**a**–**f**) and 4 (**g**–**l**) after implantation. (**e**) and (**l**) are higher magnifications of the boxed areas in (**d**) and (**k**), respectively. (**a**–**l**) Progressive bone formation occur over the entire implant. OPN-positive reactions (arrowheads) are observed in the bone-implant interface at the site of bone formation, and TRAP-positive reactions are observed in the new bone. (**l**) is the higher magnification of the boxed area in (**k**). Arrows, direct osteogenesis; B, bone; IS, implant space. Scale bars = (**d**) 500 μm, (**g**) 250 μm, (**a**,**c**,**j**,**k**) 100 μm, and (**b**,**e**,**f**,**h**,**i**,**l**) 50 μm.

**Figure 4 ijms-24-03124-f004:**
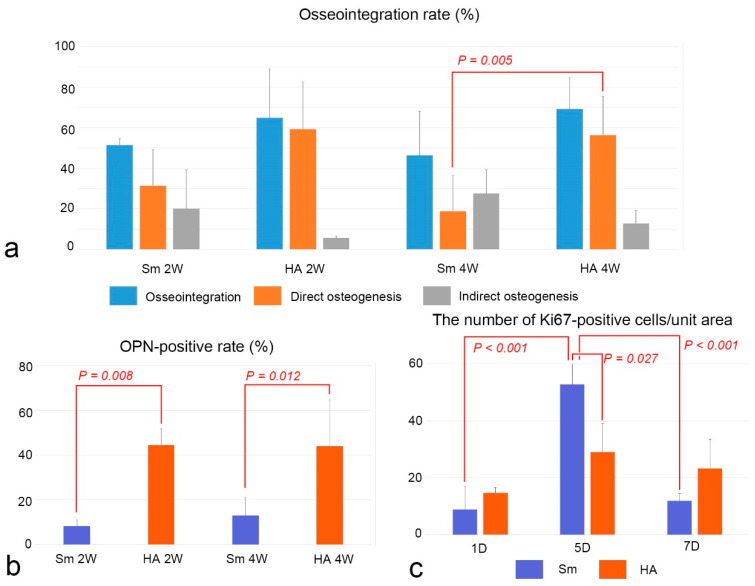
Osseointegration rate (**a**), OPN-positive rate (**b**), and the number of cell proliferation (**c**) in the HA and Sm groups. (**a**) The osseointegration rates between two groups are compared using the two-tailed Student’s *t*-test. There is a significant difference in direct osteogenesis between the HA and Sm groups at week 4. (**b**) The number of OPN-positive rates between the two groups are compared using the two-tailed Student’s *t*-test. The OPN-positive rate in the HA group is significantly higher than the Sm group at weeks 2 and 4. (**c**) The number of Ki67-positive cells among different stages after implantation is compared using one-way analysis of variance (ANOVA) followed by the Bonferroni test for multiple comparisons and the number of Ki67-positive cells between two groups are compared using the two-tailed Student’s *t*-test. Cell proliferative activity peaks at postoperative day 5 in both the HA and Sm groups and is significantly higher in the Sm group than the HA group at day 5. The numbers are the mean + standard deviation (SD).

**Figure 5 ijms-24-03124-f005:**
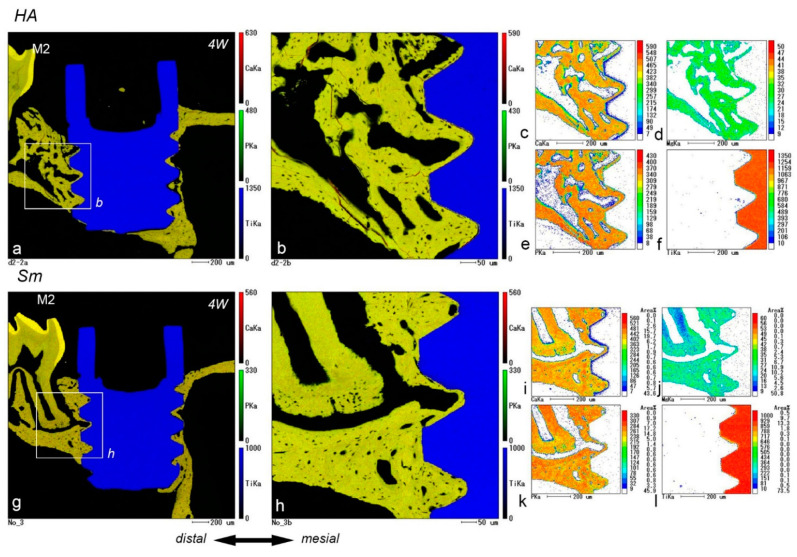
Sagittal views of the tissues surrounding the implants in the HA (**a**–**f**) and Sm groups (**g**–**l**) at week 4 after implantation obtained by EPMA procedures. (**b**) and (**h**) are higher magnifications of the boxed areas in (**a**) and (**g**), respectively. (**a**,**b**,**g**,**h**) Direct contact between the implant surface and surrounding bone is completed, although certain areas are not covered with bone. (**c**–**f**,**i**–**l**) The calcium (**c**,**l**), phosphorus (**e**,**k**), and magnesium concentrations (**d**,**j**) in the surrounding bone do not appear to be different from those of the preexisting bone. Low calcium levels are observed on the surface of the implant (**f**,**l**). IS, implant space. Scale bars = (**a**,**c**–**g**,**i**–**l**) 200 μm, and (**b**,**h**) 50 μm.

**Figure 6 ijms-24-03124-f006:**
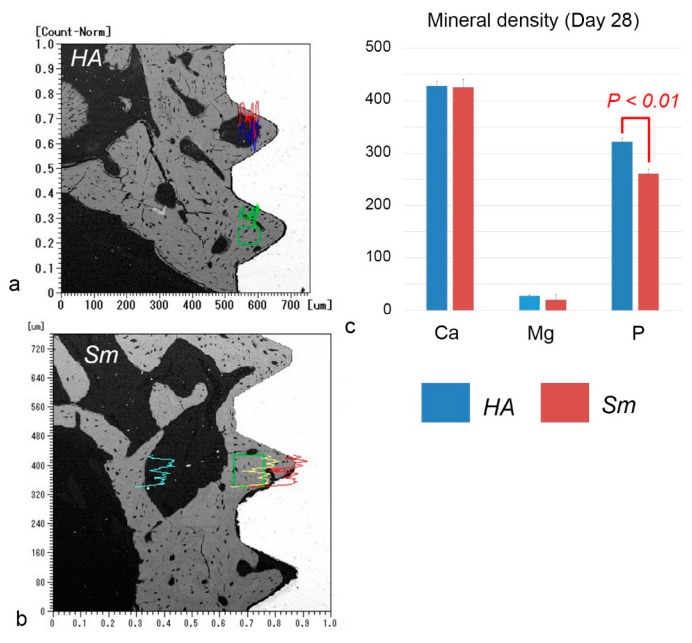
Quantitative analyses of the element densities of calcium, magnesium, and phosphorus in the surrounding bone demonstrate a significant difference in the phosphorous levels between the HA and Sm groups. The boxes in (**a**,**b**) indicate the areas analyzed for the element density. (**a**,**b**) Back scattered electron images. The line graphs colored with red, green, and blue or yellow indicate the mineral density of calcium, magnesium, and phosphorus, respectively. (**c**) The phosphorus level is significantly higher in the HA group for mineral density at postoperative week 4.

**Figure 7 ijms-24-03124-f007:**
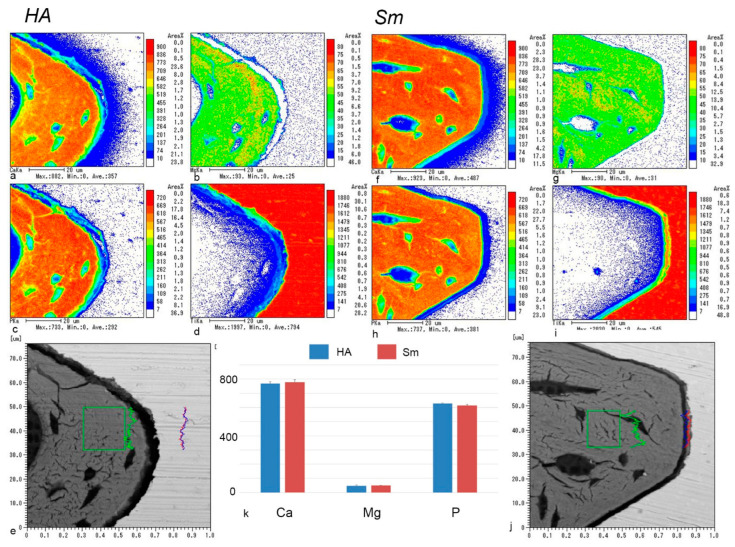
Sagittal views of the tissues surrounding implants in the HA (**a**–**e**) and Sm groups (**f**–**j**) and the element densities (**k**) at week 4 after implantation obtained by EPMA procedures. (**a**–**d**,**f**–**i**) The calcium (**a**,**f**), magnesium (**b**,**g**), and phosphorus (**c**,**h**) in the surrounding bone are similar to those of the preexisting bone. Low calcium levels are observed on the surface of the titanium (**d**,**i**) implant. (**k**) Quantitative analyses of the element densities of calcium, magnesium, and phosphorus in the surrounding bone demonstrate no significant difference in the element densities between the HA and Sm groups. The boxes in (**e**,**j**) indicate the areas analyzed for the element density. (**e**,**j**) Back scattered electron images. The line graphs colored with red, green, and blue indicate the mineral density of calcium, magnesium, and phosphorus, respectively. Scale bars = (**a**–**d**,**f**–**i**) 20 μm.

**Figure 8 ijms-24-03124-f008:**
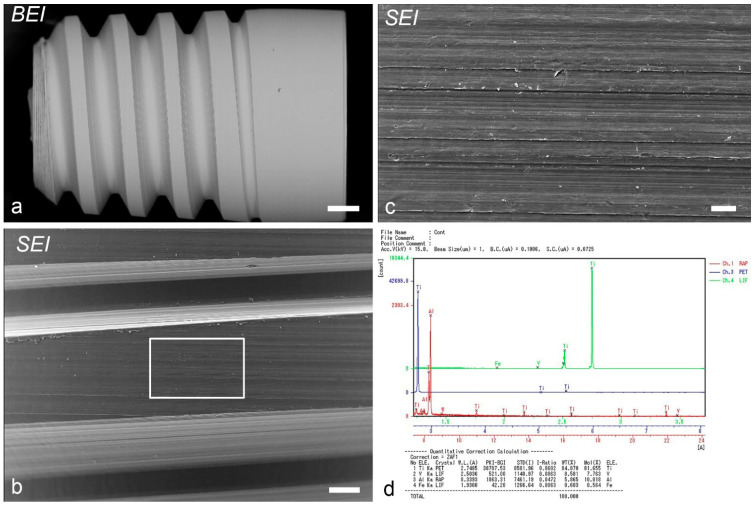
The surface morphology of a non-blasted implant (**a**–**c**) and the percentage weights of each element on the implant surface (**d**) as determined by EPMA. (**a**) A backscattered electron image of the implant. (**b**,**c**) A secondary electron image of the implant. (**c**) Higher magnification of the *boxed area* in (**b**). (**d**) Quantitative data of minerals on the implant surface. Scale bars = (**a**) 200 μm, (**b**) 50 μm, and (**c**) 10 μm.

**Table 1 ijms-24-03124-t001:** Number of animals for histological and immunohistochemical analyses for Ki67 and OPN and TRAP histochemistry.

Group	Method	Day 1	Day 5	Week 1	Week 2	Week 4	Total
HA	Histological section	3 ^1^	4 ^1^	4 ^1^	3 ^1^	6 (4 ^1^)	20 (18 ^1^)
	Ki67	(3 ^1^)	(4 ^1^)	(4 ^1^)		-	(11 ^1^)
	OPN	(3 ^1^)	(4 ^1^)	(4 ^1^)	(3 ^1^)	(6 (4 ^1^))	(20 ^1^)
	TRAP	(3 ^1^)	(4 ^1^)	(4 ^1^)	(3 ^1^)	(4 ^1^)	(18 ^1^)
	EPMA					3 ^1^	3 ^1^
Sm	Histological section	3	3	4	4	6	20
	Ki67	(3)	(3)	(4)		-	(10)
	OPN	(3)	(3)	(4)	(4)	(6)	(20)
	TRAP	(3)	(3)	(4)	(4)	(6)	(20)
	EPMA					3	3
Total		6	7	8	7	18	46

^1^ These samples were used in a previous study [5].

## Data Availability

All data generated or analyzed during this study are included in this published article.

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
