# Peer review of "Effect of Hydroxyapatite/β-Tricalcium Phosphate on Osseointegration after Implantation into Mouse Maxilla"

_ijms, 2023, doi:10.3390/ijms24043124_

Round 1

Reviewer 1 Report

Dear authors 

Analysis by paper partitions:

  1 - Introduction: it must be reformed in the content and in the writing of the general part to review the syntax of the topic  

2- Discussion : deepen in the discussion the various therapies that act in the regulation of osteoclastogenesis and osteoblastogenesis in the problems of fractures, use the following papers (optional) :

PMID: 32102398 ; PMID: 29516238 ; PMID: 29924137

  3 - Check the bibliographic entries throughout the text, some of which are non-compliant.  

4 - Review the English grammar and in particular the applied scientific English: in particular, the verb tenses and the syntax in the discussion.

Author Response

To reviewer #1

         Thank you for your kind suggestions. The points of improvement are as follows.

  1. Introduction: it must be reformed in the content and in the writing of the general part to review the syntax of the topic.

We understand the reviewer’s notion. We improved the manuscript according to the reviewer’s suggestion.

  1. Discussion : deepen in the discussion the various therapies that act in the regulation of osteoclastogenesis and osteoblastogenesis in the problems of fractures, use the following papers (optional) :
    PMID: 32102398 ; PMID: 29516238 ; PMID: 29924137

We understand the reviewer’s notion. We improved the manuscript according to the reviewer’s suggestion.

  1. Check the bibliographic entries throughout the text, some of which are non-compliant.

We totally agree with the reviewer’s notion. We carefully improved the manuscript according to the reviewer’s suggestion.

  1. Review the English grammar and in particular the applied scientific English: in particular, the verb tenses and the syntax in the discussion.

We agree with the reviewer’s notion. Although our manuscript has been edited by Enago (www.enago.jp (accessed on Jan 7, 2023)), the editing fails to correct the English grammar including the verb tenses. We carefully improved the manuscript according to the reviewer’s suggestion.

Reviewer 2 Report

The contents of the paper are interesting but the figures are not well organized.

Fig.1-2:why authors did not compare HA and SM samples also for OPN and TRAP expression? And why the authors didn't organize the figures following the same scheme?

How authors did determine osseointegration rate percentages as well as the rates of immediately placed implants?It should be explained in MM section

Fig.5 here it is stated that Ca,Mg,Ph in the surrounding bone are similar to those of the preexisting bone:how did they quantify and compare these levels ? It is not clear

Author Response

To reviewer #2

         Thank you for your kind suggestions. The points of improvement are as follows.

  1. The contents of the paper are interesting but the figures are not well organized.
    1-2:why authors did not compare HA and SM samples also for OPN and TRAP expression? And why the authors didn't organize the figures following the same scheme?

We totally agree with the reviewer’s notion. We improved the manuscript according to the reviewer’s suggestion.

  1. How authors did determine osseointegration rate percentages as well as the rates of immediately placed implants? It should be explained in MM section

We totally agree with the reviewer’s notion. We improved the manuscript according to the reviewer’s suggestion: “The percentage of osseointegration and OPN-positive perimeters in the total perimeter of the bone-implant interface was calculated using a software (Image J 1.45s; National Institutes of Health, Bethesda, MD). The direct and indirect osteogeneses were determined in the histological sections: the direct osteogenesis showed the direct bone deposition on the implant surface, whereas the soft tissue intervened at the bone-implant surface in the indirect osteogenesis (Lines 420-427).”

  1. 5 here it is stated that Ca, Mg, Ph in the surrounding bone are similar to those of the preexisting bone: how did they quantify and compare these levels ? It is not clear

We totally agree with the reviewer’s notion. We improved the manuscript according to the reviewer’s notion: “The calcium (c, l), phosphorus (e, k), and magnesium (d, j) concentrations in the surrounding bone are similar todo not appear to be different from those of the preexisting bone (Lines 236-238).”
